# Recent Advances in Electrospun Nanofiber-Based Strategies for Diabetic Wound Healing Application

**DOI:** 10.3390/pharmaceutics15092285

**Published:** 2023-09-05

**Authors:** Kun Li, Zhijun Zhu, Yanling Zhai, Shaojuan Chen

**Affiliations:** 1College of Textile & Clothing, Qingdao University, Qingdao 266071, China; qdlkun@126.com; 2College of Chemistry & Chemical Engineering, Qingdao University, Qingdao 266071, China; zhuzhijun@qdu.edu.cn (Z.Z.); zhaiyanling@qdu.edu.cn (Y.Z.)

**Keywords:** diabetic, diabetic ulcer, chronic, electrospinning nanofibers

## Abstract

Diabetic ulcers are the second largest complication caused by diabetes mellitus. A great number of factors, including hyperchromic inflammation, susceptible microbial infection, inferior vascularization, the large accumulation of free radicals, and other poor healing-promoting microenvironments hold back the healing process of chronic diabetic ulcer in clinics. With the increasing clinical cases of diabetic ulcers worldwide, the design and development of advanced wound dressings are urgently required to accelerate the treatment of skin wounds caused by diabetic complications. Electrospinning technology has been recognized as a simple, versatile, and cost-reasonable strategy to fabricate dressing materials composed of nanofibers, which possess excellent extracellular matrix (ECM)-mimicking morphology, structure, and biological functions. The electrospinning-based nanofibrous dressings have been widely demonstrated to promote the adhesion, migration, and proliferation of dermal fibroblasts, and further accelerate the wound healing process compared with some other dressing types like traditional cotton gauze and medical sponges, etc. Moreover, the electrospun nanofibers are commonly harvested in the structure of nonwoven-like mats, which possess small pore sizes but high porosity, resulting in great microbial barrier performance as well as excellent moisture and air permeable properties. They also serve as good carriers to load various bioactive agents and/or even living cells, which further impart the electrospinning-based dressings with predetermined biological functions and even multiple functions to significantly improve the healing outcomes of different chronic skin wounds while dramatically shortening the treatment procedure. All these outstanding characteristics have made electrospun nanofibrous dressings one of the most promising dressing candidates for the treatment of chronic diabetic ulcers. This review starts with a brief introduction to diabetic ulcer and the electrospinning process, and then provides a detailed introduction to recent advances in electrospinning-based strategies for the treatment of diabetic wounds. Importantly, the synergetic application of combining electrospinning with bioactive ingredients and/or cell therapy was highlighted. The review also discussed the advantages of hydrogel dressings by using electrospun nanofibers. At the end of the review, the challenge and prospects of electrospinning-based strategies for the treatment of diabetic wounds are discussed in depth.

## 1. Introduction

Over 500 million patients are currently suffering from diabetes mellitus (DM), which is one of the most common chronic diseases caused by the disorder of glucose metabolism [1]. Diabetic ulcers (DUs) are one of the main complications originating from DM, and it is estimated that approximately 20% of DM patients are accompanied by chronic diabetic ulcers with a high recurrence rate [2]. Untreated DUs can lead to tissue degeneration, death, and even amputation on the wound sites [3]. The clinical treatment of DUs is extremely costly, which most families can hardly afford [4]. Even worse, diabetic research and clinical studies have shown that the prevalence of the DU market is estimated to increase to around USD 4 billion by 2027 [5,6]. Therefore, designing and developing advanced and cost-effective treatment strategies that could significantly improve the healing outcomes and shorten the healing period of chronic diseases and hard-to-heal DUs are urgently required [5,7].

According to the reports from the American Medical Association, one of the effective treatment strategies for chronic DUs is a combined application of adopting multifunctional wound dressings and taking hypoglycemic drugs [8]. Wound dressing has been recognized as an essential medical biomaterial for helping the healing of chronic wounds in clinics [9]. Most recently, the remarkable progress in science and technology has spawned the generation of a number of innovative wound dressings, including hydrocolloids [10], hydrogels [11], porous films [12], textiles [13], 3D printed scaffolds [14], and nanofibrous dressings [13]. Although all these advanced dressing types have been demonstrated to exhibit obviously enhanced healing-promoting characteristics, compared to those traditional wound dressing materials, like gauzes, tulles, and bandages, nanofiber-based dressing materials have raised extra levels of concerns as promising dressing alternatives for a better treatment of DUs [7,15,16].

To date, electrospinning has been deemed as one of the simplest, most promising and feasible methods for fabricating nanofiber-constructed yarns, which have great benefits in serving as dressing materials for wound treatment applications [17]. Firstly, the electrospinning technique can generate fibers with diameters ranging from several to several hundreds of nanometers, which are dramatically thinner than the fibers (the diameter usually > 10 μm) generated from traditional spinning methods like wet spinning, dry spinning, and melt spinning [18]. Therefore, the electrospun nanofiber can largely replicate the fiber scale, structure, and biological performances of the collagen fibers exhibited in native skin extracellular matrix (ECM) [19], resulting in obviously enhanced biological activities and functions compared with the traditional microfibers. Recently, conductive polymers and their mixtures with other polymers were electrospun into nanofibers for tissue engineering usage; however, the nanofiber mats exhibit a substantially poorly mechanical characteristic. In order to address the issue, fabricating polymers-based nanofiber yarns (NYs) via a modified electrospinning technique has attracted researchers’ attention [20]. Specifically, the small fiber diameter, large specific surface area, and excellent ECM-mimicking morphology have been widely demonstrated to promote cell behaviors, including adhesion, migration, proliferation, and even differentiation, as well as the ECM secretion and remodeling. Secondly, the electrospinning NYs are commonly harvested in the structure of a nonwoven-like mat with small pore size but high porosity, which can serve as a natural barrier to effectively protest against the infection of external pathogenic microorganisms, but in the meanwhile ensure the normal exchange of moisture and air on the wound bed [21]. Thirdly, the electrospinning technique has high polymeric feasibility, and more than one hundred different polymers, originating from both natural and synthetic sources, have been successfully electrospun nanofibers [22]. Importantly, several different polymers can be mixed as a blend for electrospinning to satisfy the requirements of different application scenarios. Electrospinning NYs can also be utilized as effective carriers to load drugs, bioactive agents, and even living cells, and further impart the electrospinning-based dressings with some predetermined biological functions, such as hemostasis, antibacterial, antioxidation, inflammatory regulation, angiogenesis, and so on, which are of significant importance for the treatment of difficult-to-heal DUs.

In the last two decades, some dramatic advancements have been achieved in the field of designing electrospinning-based strategies for chronic wound healing applications, mainly including the selection of polymeric components, the design of dressing structure, appropriate modification, bio-functionalization, etc. This review pays special attention to the recent electrospun nanofiber dressings and enhanced hydrogel nanofiber dressing, the direction for the design, and the development of innovative electrospinning-based dressing for the treatment of chronic disease (Figure 1). The key to treating diabetic ulcer is, firstly, understanding the pathology of the diabetic ulcer, and knowing the process of wound healing. This review firstly introduces the background DUs, and the normal wound healing process, as well as the as-referred factors that hold back the healing of chronic DUs. Then, an overview of the basic information of the electrospinning process and mechanisms is introduced, and some innovative electrospinning strategies like emulsion electrospinning, coaxial electrospinning, and triaxial electrospinning are highlighted. After that, this review summarizes the selection of multi-component polymers for electrospinning and some innovative strategies like loading various bioactive agents, incorporating living cells for DU treatment applications, and building bridges between drugs and wound sites. Importantly, the innovative design of composite dressing materials by utilizing electrospun nanofibers as enhanced elements of hydrogels are also discussed. Finally, this review discusses some current challenges and prospects of electrospinning-based strategies for hard-to-heal DU treatment in clinics.

## 2. Diabetic Ulcers (DUs)

There are two major types of DM [23,24,25]. Type-1 diabetes alters the blood circulation of different blood vessels and delays oxygen delivery to the diabetic wound site [26,27,28]. Type-2 DM is associated with chronic foot ulcers and is responsible for a high mortality rate, which requires a localized drug delivery system (DDS) that delivers drugs to help wound closure [29]. Diabetes is characterized by an increase in blood glucose [30], which results in diseases such as diabetic ulcers, leg ulcers, and buttock pressure ulcers [31,32]. Diabetic ulcers may appear because of three predisposing factors: abnormal cellular/inflammatory pathways, peripheral neuropathy, and vascular disease [33]. Peripheral arterial disease, a consequence of diabetes, occurs from fat deposits and decreased blood flow to the legs and feet [34], which then creates a DU wound.

After a better understanding of the pathology of the DUs, to cure the chronic ulcer, it is essential to understand the wound healing process. The normal healing process of skin wounds is divided into four phases: hemostasis, inflammation, proliferation, and remodeling [35,36], and these phases can occur all at once or overlap [37]. Hemostasis is preventing the secondary flow of blood after a blood clot has formed in the wound site [16]. The release of platelet-derived growth factor (PDGR) and transforming growth factor (TGF) occurs [38]. Once the healing is complete, the inflammatory response subsides, and neutrophils are cleared from the wound area. Afterward, in the proliferation phase, macrophages produce anti-inflammatory macrophages (M2). M2 macrophages release transforming growth factor β1 (TGF-β1) and vascular endothelial growth factor (VEGF), which induce angiogenesis and granulation tissue formation [39,40].

The high blood glucose level in the human body results in symptoms such as tissue hypoxia, for instance, a slow response in tissue [41], a weak blood supply, and increased chances of receiving infections [42,43]. Diabetic wound healing is delayed because pathogens increase the changes of tissue hypoxia by reducing the process of angiogenesis and endothelial revascularization; therefore, building an environment that could mimic ECM is important [44]. Electrospun nanofibers could highly recreate the ECM environment. The hyperglycemic condition in diabetic people prevents the cellular proliferation of keratinocytes, fibroblasts, and endothelial cells, affecting re-epithelialization and neovascularization [45]. The most dysregulated phase is the inflammatory phase [46]. It has been proven that diabetes results in dysregulated processes from proinflammatory (M1) macrophages to anti-inflammatory (M2) macrophages conversion [15,47]. Thus, diabetic ulcers show deregulated angiogenesis, and inflammatory response takes longer to process because the upregulation of MMP-9 in the case of diabetes leads to the degradation of the ECM. To cure DUs, the key is to utilize the electrospinning method to mimic ECM structure [5,40,48]. Therefore, it needs to explore more effective ways to treat diabetic ulcer, such as exploring the electrospinning techniques with different polymers, adding bioactive agents to nanofiber dressings and enhancing nanofiber hydrogels.

## 3. Mechanisms and Apparatus for Electrospinning Technology

In the past two decades, electrospinning has been extensively employed to fabricate nanofibers for diabetic diseases [49,50] due to its ability to produce polymers composed of fibers with a small diameter (10 nm to 10 mm) that have a high surface-to-volume ratio, chemical versatility, mechanical properties, and porosity; moreover, it can control the morphology of the fibers [51,52]. The traditional nanofiber-generating unit setup for electrospinning includes three parts as shown in Figure 2A: a high-voltage power supply (HVPS), a conductive collector utilized to collect the deposited nanofibers, and a syringe with a needle. The high-voltage supply is used to provide a necessary driving force to generate an electric field, and the high-voltage power supply is attached to a polymeric solution as an electrode having a positive charge and adheres to a metallic collector with a negative charge [53]. The syringe is filled with the needed spinning solution with a controllable solution feeding rate, which is utilized to produce nanofibers continuously [37]. A strong electrostatic field can be formed between the spinneret and a ground collector when a high-voltage supply connects to the spinneret during the process. As voltage increases, there is a rise in electrostatic charges that collects on the surface of the liquid droplet side of the spinneret. When the electrostatic repulsion force reaches the highest amount over the polymer solution, the liquid droplet starts acting as a Taylor cone and ejects from the tip of the syringe as a Taylor cone. The polymeric jet ejected toward the ground collector and solvent might be evaporated during the process, and eventually, nanofibers are deposited on the device [54]. In contrast, during the bending instability stage of electrospinning, polymeric jets are stretched, and the force produced by the external electrical field and other factors repeatedly stretches the polymeric jet to produce fibers with diameters ranging from several to several hundred nanometers [37]. Also, according to Deitzel et al., increasing the electrostatic voltage could decrease the initial jet’s stability, which may result in a beading defect in the fibers [55]; therefore, controlling the voltage shows great effects on the morphology of the fibers. Moreover, electrospinning can also be affected by a few factors, for instance: the humidity and temperature of the room, the flow rate (µL/min), and the distance between the needle tip to the collector (cm). Therefore, it is necessary to control the fiber structure and monitor the electrospinning process while considering the above factors.

Electrospun nanofibers can be effectively adjusted and tailored by using electrospinning techniques. Based on the structure of the nozzles, electrospinning methods can be divided into different categories: blending electrospinning, emulsion electrospinning, side-by-side electrospinning, and coaxial electrospinning. As shown in Figure 2B–D [56], all the methods exhibit different potentials. Chen et al. utilized a simple one-step electrospinning technique only to fabricate fibrous sorbents that exhibit adsorption capability; the method is easy and contains good combinations of cosolvents, and the fibers can adsorb oils while repelling water, which is a very successful attempt [57]. Blend electrospinning is used for solutions containing a hybrid of two or more polymers, which can combine all the advantages of different polymers into one mixed polymer through the most common and simplest electrospinning technique. Suresh et al. identified that blending electrospinning shows good performance in mixing poly-ε-caprolactone and gelatin, but vertical orientation shows significantly higher cell viabilities than horizontal orientation [58]. The structure of emulsion electrospinning is similar when compared to blending electrospinning, and the important section is to prepare the two-phase emulsion to form core–shell electro-nanofibers [59]. Additionally, compared to both blend electrospinning and emulsion electrospinning, loading drugs into the core layer via coaxial electrospinning can reduce the burst release and increase the time of loaded drugs’ release [60]. Side-by-side electrospinning can be used to allow different solutions to flow through separate tubes [61]. Another popular electrospinning technique is coaxial electrospinning. There have been many modifications made to the coaxial electrospinning setup. In coaxial electrospinning, two needles are parallel to one another and in a core–sheath structure to produce nanofibers. The solution is in the same flow rate as it moves to the syringe. This method could also combine all the polymer merits into one solution. Yin et al. investigated using coaxial electrospinning technology to fabricate the desirable small-diameter vascular grafts for wound healing by collagen/chitosan/poly(l-lactic acid-co-ε-caprolactone) (PLCL). The result shows that heparin exhibits a sustained release even after 45 days. The test results proved an excellent cell biocompatibility and suitable mechanical properties [62]. Li et al. illustrated a novel low-cost wound dressing PLCL/Gelatin/Epigallocatechin-3-O-gallate (EGCG)/Core–shell nanofiber membrane (PGEC) for improving the low bioavailability of EGCG via coaxial electrospinning technology. The result shows that the core–shell structure enhanced the wound healing process, promoted wound regeneration, and overcame the weakness of EGCG; it had high potential for clinical application [63]. There are also a variety of collectors, which allows fibers to align in different directions. Single-nozzle electrospinning is conducted by adding a voltage across a conductive collector, while the conductive spinneret supplies the precursor solution. Han et al. illustrated that the designed collectors can change the alignment of electrospun nanofibers [64]. Aside from changes to the mechanical structure, Xu et al. introduced a battery-operated portable handheld spinning apparatus (BOEA), in which two batteries and one high voltage converter are used to replace the high voltage power supply, eliminating the need for the power plug [65]. This portable apparatus allows the spinning technique to be applied and used in more mobile applications. Although some of these innovative methods for electrospinning methods have been applied to cure diabetic ulcer, it has not been widely used, but can be utilized in future DU treatments [66].

## 4. Multi-Component Polymers in Electrospun Nanofibers for DU Treatment

Different polymers and different electrospinning methods can be selected for electrospinning nanofibers, which can exhibit different chemical or physical properties [67]. To achieve the desired successful healing of diabetic ulcers, selecting the ideal multi-component polymers and applying them with electrospinning technology are urgent requirements. The frequently employed synthetic polymers include PCL [68], PLGA [69], PVA [70,71], PLA [72,73], and PLCL [74,75,76]. Natural polymers show more bioactive and biocompatible behaviors but less mechanical properties. A lot of research has proven that the commonly applied natural polymers for DUs are chitosan (CS) [77], collagen [78], gelatin (GT) [79], zein [80], elastin [81], silk [82], alginate [83], etc.

Poly(lactic-co-glycolic acid) (PLGA) is a synthetic polymer that is often used for nanofibrous fabrication to promote wound healing. PLGA is biodegradable and biocompatible and contains excellent mechanical strength that can be hydrolyzed in aqueous solution. PLGA is a non-toxic material to humans as it leads to minimal inflammation and it can biodegrade to lactic and glycolic acid, which are biocompatible as well; most importantly, it can control drug release [84]. For instance, Liu et al. found that multi-component polymers mixed with PLGA/collagen in a HFIP solvent could help further explore cell behavior, cytocompatibility, and most importantly, help promote faster wound healing; therefore, it is possible to mix synthetic and natural polymers to achieve better results [85]. Figure 3A,B illustrate the schematic of the process.

PCL is a synthetic polyester that has also been applied in electrospinning for diabetics. It is a biodegradable and biocompatible polymer that has the quality for implantation due to its viscoelastic and bioresorbable characteristics. But its hydrophobicity prevents cell adhesion, and it is suggested to blend PCL with other hydrophilic polymers to provide the hydrophilic characteristics [88,89]. Polyvinyl alcohol (PVA) was selected to produce electrospun nanofibers, owing to its excellent fiber-forming quality, biocompatibility, biodegradability, mechanical function, and chemical resistance properties [90]. It can cross-link with other polymers to obtain a stable nanofiber for wound-healing dressing [91]. Poly(lactide-co-ε-caprolactone) (PLCL) has excellent biocompatibility, biodegradability, and mechanical properties [92]. Zhang et al. discovered a method that used a low-pressure filtration-assisted solution mixed with poly(L-lactate-caprolactone) PLCL nanofibrous/keratin hydrogel bi-layer wound dressings, and then loaded with fibroblast growth factor (FGF-2). The keratin hydrogel shows 874.09%, the elastic modulus of hydrogel shows great similarity with the human dermis. As shown in Figure 3D,E, this wound dressing shows a better wound repair effect and promotes re-epithelialization [87]. Liu et al. found that the core–sheath fiber SF/PLCL shows a promising candidate for tissue engineering scaffolds. The outcomes demonstrated that the fiber mat enabled cellular adhesion and proliferation, which could promote wound healing faster [86]. Chitosan is a biological material obtained through chitin deacetylation with excellent biocompatibility, biodegradability, and hemostasis blood clotting capability, which is an excellent candidate material for mimicking native ECM for the scaffold. Jamnongkan et al. suggested applying PVA and chitosan to fabricate electrospun nanofibers for wound healing purposes, because electrospun pure chitosan is hard, and when added with PVA, could lower the difficulty of electrospinning [72]. PLA is also a biodegradable synthetic polymer that not only has high mechanical strength, but also high biocompatibility, cell attachment, and proliferation [93]. The most prevalent biopolymer in nature after cellulose is chitosan (CS). It contains various therapeutic usages, including wound healing, and the binding of CS can promote red blood cells coagulate rapidly [94]. Gelatin is a natural polymer that has good biocompatible and hydrophilic properties, which is from the hydrolysis of collagen and contains a great amount of glycine, proline, and hydroxyproline residues [95]. However, the gelatin/water solution often shows poor electrospinning ability. To improve the spinnability, acidic aqueous solvents need to be added to the electrospinning process. Electrospun gelatin-based nanofiber could protect skin wounds from bacterial invasion and keep moisture permeability. Collagen is the most important component of natural ECM [96]. Also, collagen has great potential for cell attachment, proliferation, and differentiated function in tissue culture. Elastin is the organ of blood vessels, muscle, skin, and articular cartilage; its major function is to provide elastic recoil in extensible tissues [97]. Silk is a mutable function and chemically appealing material with attractive biological and mechanical properties. It is from amino acids’ composition and sequence, which contains a three-dimensional structural organization. It is widely used in electrospinning technology [98]. Zein is an alcohol-soluble prolamin protein that can be electrospun with different solvents, like acetic acid or aqueous methanol [99]. In addition, the hybrid electrospinning of zein and other biopolymers has been successfully electrospun. Deng et al. identified that electrospinning gelatin/zein nanofibers show good deformability, flexibility, and water resistance [100]. Alginate is widely used as scaffolds and wound healing dressing, it is an acidic liner polysaccharide composed of guluronic acid and mannuronic acid moieties [101]. It should be noticed that most of these multipolymers have already been utilized in diabetic ulcer treatment via electrospinning technology, but to obtain better results, wound dressing fabricated by multiple polymers loaded with bioactive agents through the electrospinning technique is the next recommended step. For those multipolymers that have not yet been used, they can be utilized in future diabetic ulcer treatments as well. Table 1 summarizes some representative electrospun-based multi-polymers for diabetic wound healing purposes.; some of the experiments have applied the materials to animals such as diabetic rats and rabbits. The results proved that multi-polymers help the experimental animals heal faster. Based on the table, it is well illustrated that mixing synthetic polymers and natural polymers together could reach a better healing result, which promotes cell proliferation and help wounds heal faster.

## 5. Electrospun Nanofiber Dressings Loaded with Bioactive Agents

Electrospun nanofibers have been utilized to mimic natural scaffolds for skin regeneration and wound healing; however, the result is still not satisfactory. Bacteria on the surface of the dressings could impact the procedure of the healing process. One of the reasons that healing is difficult for chronic wounds is inflammation. Based on the organism of the wound healing process, the novel strategy is to integrate the bioactive agents into dressing materials for promoting wound healing [114]. Bioactive agent delivery systems via electrospun nanofibers have been widely developed in the last decade, especially in the tissue scaffolds field. Therefore, loading with bioactive agents on electrospun nanofiber dressings or dissolving the bioactive agents and polymers in the same solvent [115] can accelerate the healing process in chronic diseases, as shown in Figure 4A [116]. Electrospun nanofibers could also deliver bioactive ingredients, antibiotics [12], antibacterial particles [117], stem cells [118], metal nanoparticles [119], and plant metabolites [120,121], which can activate cell proliferation, migration, and differentiation of damaged cell tissues. Hence, besides selecting different materials for rebuilding the natural scaffolds, electrospun nanofibers are also utilized as the delivery system for loading different bioactive agents to wound sites. Bioactive agents discussed below have demonstrated that they assist in the wound healing process. Metallic oxide nanoparticles such as ZnO, CuO, TiO_2_, and CeO_2_ have demonstrated bactericidal activity and been extensively accepted as antibacterial agents. Mei et al. introduced a wound dressing that prevents bacterial infection by applying antibacterial particles and zein/cinnamon oil through a handheld electrospinning method, and this novel dressing showed gas permeability of (76.1 ± 5.45) mm/s. The mice experiment illustrated that zein/cinnamon oil wound dressing could heal the wound in around 11 days. When CO was mixed into the solution, the fabricated zein/PEO/CO dressing had obvious inhibition zones over 5 cm for the two bacterial strains after 24 h intervals, which provided good antibacterial properties, as shown in Figure 4C [122]. Chen et al. summaries some antibacterial particles that have been loaded into electrospun nanofibers for wound healing purposes, such as lysozyme [123], silver [30], and ciprofloxacin HCI [124]. Doxycycline (DOX)-loaded nanofibers shows excellent antibacterial efficacy; it is an appealing candidate for treating DUs [125]. Shalaby et al. formulated cellulose acetate-based nanofibers loaded with silver nanoparticles as a wound dressing for the treatment of diabetic disease. The result shows that sliver nanofibers can be applied as antimicrobial wound dressing materials. Liu et al. discovered an absorbable HHA-based nanofibrous hydrogel, which was then crosslinked with Fe^3^ to adjust the inflammation microenvironment to accelerate diabetic wound healing [126].

According to the International Working Group on the Diabetic Foot (IWGDF), once the infection and bacteria are ensured, antibiotic agents should be applied onto nanofibers for diabetic treatments [127,128]. Antibiotics are essential agents in dealing with wound healing, which show quick efficacy and contain stronger bacterial abilities. Hence, it is necessary to add antibiotic agents onto electrospun nanofibers. Doxycycline hydrochloride (DCH) is one of the important antibiotic drugs that can reach an antibacterial effect. Cui et al. determined that DCH-loaded PLA-based electrospun nanofibers for chronic wound management have a huge impact on chronic wound treatment. When the ratio of DCH increased, the drug release was fast, and it fully released the drug after 2 weeks, as shown in Figure 4D [125].

Metal nanoparticles do not interact with the specific bacteria, metal nanoparticles as a bioactive agent loaded into electrospun nanofibers will improve the process for DU treatment. One of the most viable nanoparticles used in diabetics is AgNPs: AgNPs speed up the rate of wound healing due to the anti-inflammatory properties on the strains of multidrug-resistant bacteria [30]. Tian et al. researched the effect of AgNPs on wound healing properties in animal models. They found that the animal model shows positive results with Ag’s antimicrobial activity and cytokine modulation [120]. According to the finding of Lau and Barath Mani Kanth, AuNPs have anti-oxidative and antimicrobial effects on improving the diabetic wound healing process [129,130]. For the antibacterial purpose, the best candidate is copper nanoparticles (Cu NPs). Cu NPs have gained attention for curing DUs based on their antibacterial properties which target bacterial strains that are present in diabetic infections; for instance, Escherichia Coli and Staphylococcus aureus [131]. Tiwari et al. concluded that CuNPs help diabetic wound healing rate and biosynthesis and promotes faster wound healing [132]. However, the potential risks of the metals utilized are also a concern for the medical community, as metal nanoparticles such as silver may develop harmful reactions such as liver function impairment, pain, etc. [133]. In addition, more than one bioactive agent can be applied to electrospun nanofibers to achieve a better result for the healing procedure. For instance, Jafari et al. designed double-layer nanofibers, which contained PCL/gelatin loaded with amoxicillin on one side of the dressing and ZnO nanoparticles on the other layer. The drug exhibited a longer release time for amoxicillin, up to 144 h (tested time) in vivo, since the layer loaded with ZnO nanoparticles easily forms a barrier to drug release. The potency of scaffolds for hindering bacterial growth alongside the results of a cytotoxicity evaluation revealed that it can help cell proliferation. In the end, the experiment showed that the drug promoted in vivo angiogenesis, so the scaffold became a promising candidate for wound treatment (Figure 4E) [134]. Cam et al. discovered a novel material for diabetic wound treatment that indicates the non-toxicity and good biocompatibility of nanofibers: bacterial cellulose–gelatin nanofibers loaded with metformin [135], which promotes faster wound healing as well as healing of the wound site.

**Figure 4 pharmaceutics-15-02285-f004:**
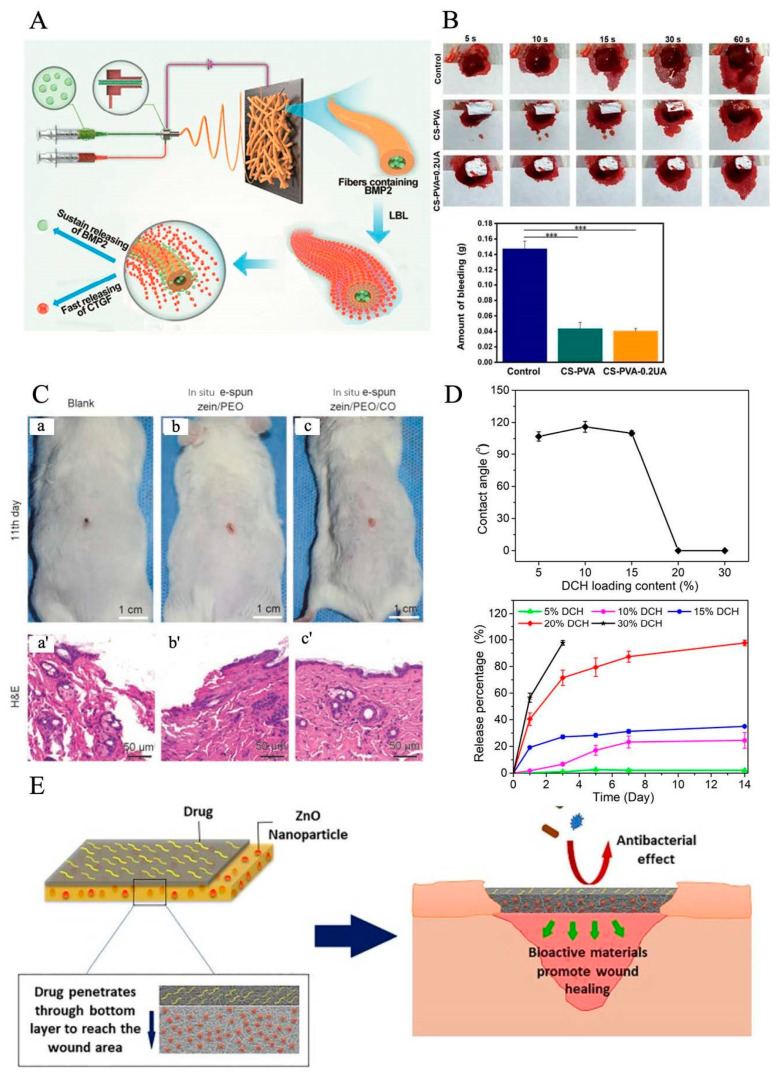
(**A**) Illustration of sustained release of BMP2 and transient release of CTGF, which could facilitate these proteins to play their respective roles at different stages of bone healing. (**B**) The actual photographs of mouse hepatic hemorrhage using the CS-PVA nanofiber mat and the CS-PVA-0.2UA nanofiber mat as hemostatic materials over 60 s. Control stands for the mouse hepatic hemorrhage without any treatments. The image below is the corresponding analysis of bleeding amount in three different groups after 60 s of bleeding. Statistical analysis was carried out using ANOVA with Scheff´e post-hoc tests. When *p* < 0.05, there was significant difference between two groups. When *p* < 0.001, there was not significant difference (***). Reprinted with permission from [136]. (**C**) Efficiency of in situ deposition zein/PEO and zein/PEO/CO dressings on Kunming mice wounds’ model on the 11th day after injury. The wound closure of blank group without any management (**a**), in situ depositionzein/PEO dressing control (**b**), and in situ deposition zein/PEO/CO dressing (**c**). (**a**′–**c**′) HE staining histological images of the above wound closure cases. Reprinted with permission from [122]. (**D**) Wettability of DCH/PLA nanofiber mats and DCH release behavior. (**Top image**) Dependence of contact angle of DCH/PLA nanofiber mats on DCH loading content. (**Bottom image**) Dependence of DCH release behavior on its content in DCH/PLA nanofibers (*n* = 3) during the experimental period of 14 days. Reprinted with permission from [125]. (**E**) Schematic representation of the antibacterial and bioactivity mechanism of nanofibers in the wound area. Reprinted with permission from [134].

Plant extracts contain a lot of bioactive agents, such as essential oil (EOs) and calendula officinalis (CO) [137]; for example, Golchin et al. introduced a novel electrospun nanofiber via cur-incorporated chitosan/PVA/Carbopol/polycaprolactone nanofiber that effectively heals wounds, because chitosan and PVA have the characteristics of promoting healing. Ursolic acid (UA) is a bioactive extract from Chinese herbal plants. Lv et al. innovated electrospun nanofibers made from a blend of chitosan and polyvinyl alcohol (PVA) and loaded with UA for diabetic wound treatment. After 18 days of treatment observation, the data show that about 99.8% of the diabetic wound area was healed. The hemostatic effect of mouse hepatic hemorrhage models shows CS-PVA-0.2UA was 0.0397 g less than the CS-PVA group; this shows that UA and CS could effectively promote the agglutination of red blood cells and platelet aggregation. This agent significantly promoted the regeneration of diabetic ulcers (Figure 4B) [136]. Almasian et al. fabricated PU-carboxymethylcellulose nanofibers loaded with Malva sylvestris for diabetic wound treatment. The result shows that on day 14, the traditional gauze had a 32.1 ± 0.2% healing rate, but the plant extract-loaded nanofiber dressing’s healing rate was 95.11 ± 0.2% [138]. Peng et al. [139] discovered a sesquiterpene lactone compound extracted from Artemisia annua (ART) as a bioactive agent, electrospun with PLGA/SF membranes to reach antibacterial properties for wound healing purposes. Antimicrobial peptides have better characteristics for antibacterial function because they are a part of innate epithelial chemical shields with advantages, including broad-spectrum activity. Since their degradation ingredients are natural amino acids, it is safe for silver to be applied for human body care. Xie et al. designed an in situ forming hydrogel (IFBH) system composed of a newly developed combination of poly(ethylene glycol) maleate citrate (PEGMC) and poly(ethylene glycol) diacrylate (PEGDA). Their studies show that the in situ forming antimicrobial biodegradable hydrogel system is a useful candidate for wound treatment [140]. In addition, living cells can also be employed for promoting wound treatment applications. Fu et al. discovered a bioactive agent: human urine-derived stem cells (hUSCs), which could secrete multiple growth factors such as vascular endothelial growth factor (VEGF), etc., with the hUSCs-seeded electrospun PCL/gelatin nanofiber dressings, which promote wound healing [141]. Table 2 illustrates some popular bioactive agents incorporated into nanofibers for improving wound healing, for instance: ibuprofen, berberine, curcumin, dimethyl sulfoxide, etc. The experiments have been applied to animal models such as diabetic mice, STZ-infected diabetic mouse, etc.

## 6. Electrospun Nanofiber/Hydrogel Composite Dressing

Electrospun nanofibrous dressings can be utilized as barriers for bacteria, but the wound sites need to be moist. Therefore, combined electrospun nanofibers with enhanced hydrogel could be a great choice. Hydrogel materials with three-dimensional network structures have also been widely explored as advanced wound dressings for promoting the healing process of DUs [157,158]. Hydrogel-based dressings that are commonly fabricated from biopolymers like collagen (Col) [159], alginate (ALG) [101], gelatin (Gel) [160], silk fibroin (SF) [161], hyaluronic acid (HA) [162], chitosan (CS) [91], their derivatives and blends, etc., can mimic the physical and chemical performances of the native ECM components to a large extent, which can assuredly provide a beneficial microenvironment for the healing of DUs [163]. Except for their high biocompatibility and biodegradation, they have excellent hydrophilicity and water-retention capacity, which can effectively absorb wound exudates, and also offer ideal moisture management on the wound bed [164]. Like electrospinning nanofibers, hydrogels can be utilized to encapsulate drugs, bioactive reagents, and even living cells that can be designed to solve targeted pathological problems and further improve the therapeutic outcomes of complex DUs [36,41,165].

Compared with those conventional hydrogel dressings, the recent development and application of composite dressings consisting of electrospun nanofibers and hydrogels, also named as “electrospun nanofiber/hydrogel composite dressings”, have aroused much more attention for the treatment of DUs [166]. The largest characteristics of these composite dressings are their capacities, which combine the advantages of both hydrogels and electrospun nanofibers [167]. Ilomuanya et al. firstly fabricated electrospun PLA nanofiber mats with fiber diameters ranging from 300 to 490 nm and porosities ranging from 63.90 to 79.44%, and then loaded as-formulated HA-valsartan hydrogel on the electrospun PLA nanofiber mats in order to generate nanofiber/hydrogel composite dressings [168]. Animal studies showed that composite dressings significantly promote the healing of DUs by effectively reducing the inflammation response, improving angiogenesis and re-epithelization on the wound site [169]. Augustine et al. designed a novel composite dressing by combining electrospun PHBV nanofiber mats with gelatin–methacryloyl (GelMA) hydrogel, as shown in Figure 5 [170]. The PHBV nanofiber mats were demonstrated to possess mechanically stable structures to provide great barrier properties against external microbes, while the GelMA hydrogel could effectively absorb wound exudates and offer moisture management on the wound bed. In addition, epidermal growth factor (EGF) was encapsulated into GelMA hydrogel for providing angiogenesis-promoting biofunction. The results from in vitro cell characterization found that the PHBV nanofiber/GelMA-EGF hydrogel composite dressings could dramatically promote the cell activities like the migration and proliferation of multiple different cells, including fibroblasts, endothelial cells, and keratinocytes. The animal studies further demonstrated the enhanced healing-promoting capacities of as-prepared composite dressings for diabetic ulcer treatment applications.

Most of the electrospun nanofiber/hydrogel composite dressings are fabricated from electrospun mats with a randomly oriented fibrous pattern, which lacks the necessary cell guidance capacities [171]. To address this issue, some research works have fabricated electrospun mats with predetermined nanofiber patterns, which can act as guidance channels to control and direct cell migration when the electrospun mats are integrated with hydrogels. For instance, Wu et al. fabricated one type of novel electrospun mats with radially oriented fibrous patterns from a polymeric blend of methacrylated gelatin (MeGel) and PLLA, named MeGel/PLLA radially oriented nanofiber mats (RNMs) [172]. They also fabricated electrospun MeGel/PLLA haphazardly oriented nanofiber mats (HNMs) and MeGel/PLLA uniaxially oriented nanofiber mats (UNMs), and further compared how the three different nanofiber orientation patterns influence the cell behaviors. The results from cell characterization showed that the RNMs could provide excellent recruitment and guidance capacities, resulting in significantly enhanced cell viability, migration, proliferation, and alignment of human dermal fibroblasts compared with the traditional HNMs and UNMs. The animal studies based on mice acute skin wound models further confirmed the outstanding recruitment ability of RNMs, whereby the radially aligned fibers are akin to motorways that direct the autologous cells migrating from the periphery to the mat center in an obviously faster manner compared with both HNMs and UNMs. More interestingly, the electrospun MeGel/PLLA RNMs were further utilized to integrate with Salvia miltiorrhiza Bunge-Radix Puerariae herbal compound (SRHC)-loaded MeGel hydrogels to generate the novel bi-layered composite dressings (Figure 6), which were demonstrated to greatly reduce the inflammation and dramatically promote the angiogenesis and re-epithelialization of full-thickness DUs. Importantly, the composite dressings with a 10% SRHC loading rate could largely shorten the healing time, and the healing ratio of mice full-thickness DUs could reach 97.4 ± 2.8% after 18 days of treatment. In another study reported by Zhong et al., a near-field electrospinning strategy was utilized to generate electrospun PCL nanofiber mats with grid-like morphology, and the fibers with controllable sequencing in the grid mats could serve as guidance channels to effectively regulate the cell adhesion and migration [173]. The electrospun PCL grid mats were further incorporated into deferoxamine-loaded GelMA hydrogels to construct one new type of electrospun nanofiber/hydrogel composite dressing, which was found to effectively improve the angiogenesis and antioxidation, further resulting in a high-quality regeneration of damaged skin tissues in diabetic mice. To sum up, hydrogel-based dressings can promote cell adhesion and proliferation faster and repair diabetic ulcers in a more advanced way.

## 7. Conclusions

Diabetic ulcers are one of the main diabetes consequences since they can cause significant harm, necessitate amputation in certain circumstances, and still cost large sums of money to find the perfect solutions to repair the DU. Controlling the wound’s growth is essential for avoiding infections and accelerating the healing process. Researchers are paying more attention to the pathogenesis and pathophysiology of chronic diseases like diabetic ulcers in view of the information provided above. Medicines or bioactive agents that can be delivered by electrospun nanofibers show great results to the wound sites. So far, electrospinning technology is the best candidate for curing DUs because electrospinning nanofibers have a high surface area-to-volume ratio and diameter size as well as a structure similar to ECM. In this review, the electrospun nanofibers used as wound dressings for DUs have been analyzed, and the multi-component polymers used as wound dressings for DUs have been introduced. This research also suggested electrospinning loaded with bioactive chemicals to aid tissue regeneration and speed up healing to better facilitate DU treatment. In order to restore the harm caused by DUs, the bioactive agent is coupled with electrospun nanofibers for DU treatment. A variety of bioactive agents, including antibiotics, metal nanoparticles, and antimicrobial peptides, have been loaded onto electrospun nanofibers to improve the healing process. Although many studies and research have already shown that nanofibers with loaded bioactive agents can help DU treatment, there is still a lot to explore, discover, and invent. Finally, this review reported enhanced hydrogel dressings for the further improvement of DU treatment, and well summarized the advances in the fabrication and application of hydrogel dressings for DU treatment, and the summary can help researchers to understand more about the functions of hydrogel dressings.

To eliminate severe complications, prevent lower limb amputation, and lower overall morbidity and mortality rates, the therapeutic care of diabetic foot ulcers necessitates a comprehensive, interdisciplinary approach. The current approach can yield the best results, but entails sufficient preventive measures (patient education, stringent blood glucose control, self-care of the foot) and can prompt the initiation of specific treatment, such as wound correct debridement, offloading, the topical use of advanced dressings, negative pressure wound therapy, and surgical treatment. In the end, resolving the inflammatory situation has been a goal for diabetic ulcer and wound care treatment. Electrospinning technology has proven to be one of the best candidates for wound-healing dressings; however, its mechanical feature requires high voltage as it is difficult to scale up and be used at an industrial level. Therefore, other flexible and easy techniques have been considered as well. DU research has been facing some significant obstacles all this while, but it might herald a better future if we quickly adapt the different aspects as we progress into further research.

## Figures and Tables

**Figure 1 pharmaceutics-15-02285-f001:**
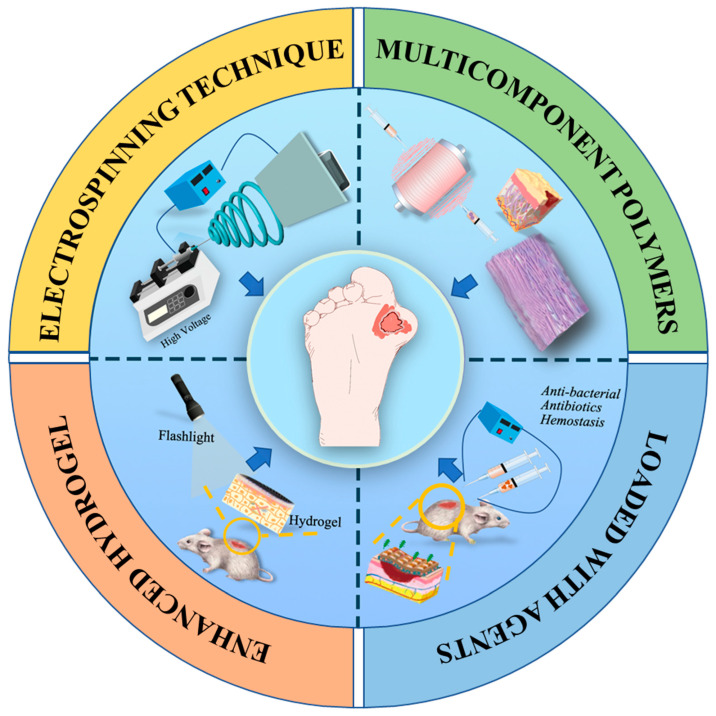
Illustration of various electrospinning-based strategies for DU treatment applications.

**Figure 2 pharmaceutics-15-02285-f002:**
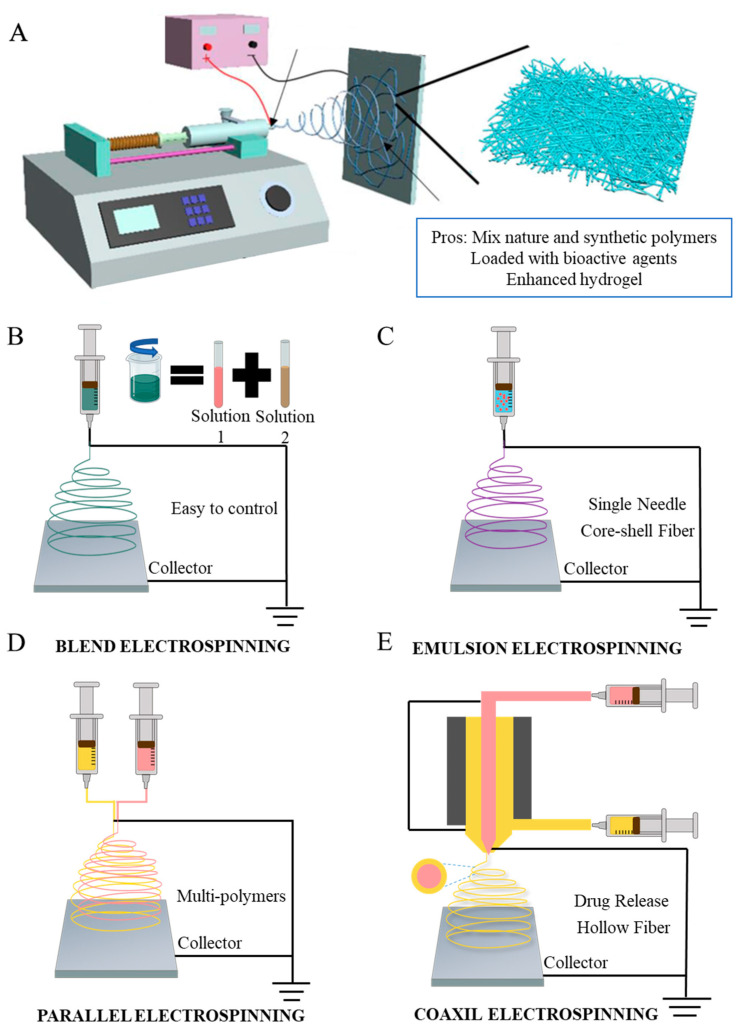
(**A**) The schematic setup of vertical conventional electrospinning. Schematic illustration of four electrospinning methods. (**B**) Blending electrospinning. (**C**) Emulsion electrospinning. (**D**) Side-by-side electrospinning. (**E**) Coaxial electrospinning.

**Figure 3 pharmaceutics-15-02285-f003:**
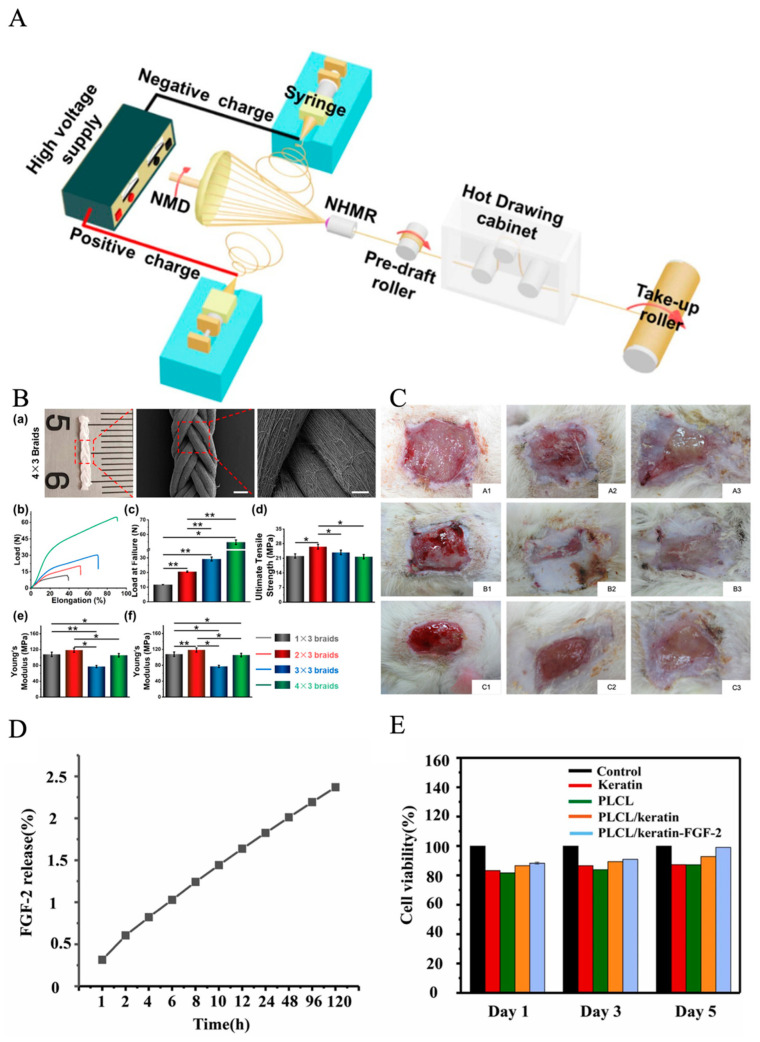
(**A**) Schematic illustration of the modified electrospinning system for generating PLLA NYs without/with hot drawing, reprinted with permission from [86]. (**B**) Mechanical properties of PLLA. (**a**) Photograph and SEM images of nanofibrous braids made of 12 PLLA NYs, named as 4 × 3 braids. Scale bars: 0.5 mm for left image and 100 µm for the right image. Uniaxial mechanical testing of the as-fabricated four nanofibrous braids using different yarn numbers, i.e., 1 × 3 braids, 2 × 3 braids, 3 × 3 braids, and 4 × 3 braids; (**b**) representative load–elongation curves; (**c**) load at failure; (**d**) ultimate tensile strength; (**e**) Young’s modulus; (**f**) elongation at failure. (*n* = 3; * *p* < 0.05, ** *p* < 0.001) [86]. (**C**) Appearance of wound healings at 1, 2, and 3 weeks after grafting (**A1**–**A3**) gauze group, (**B1**–**B3**) nanofibers group, and (**C1**–**C3**) commercial dressing group, reprinted with permission from [85]. (**D**) Results of CCK-8 assays for cell viability, reprinted with permission from [87]. (**E**) Quantitative data of the hemolytic ratio. Data are mean ± SD, *n* = 3, (*t*-test, two-tailed). Reprinted with permission from [87].

**Figure 5 pharmaceutics-15-02285-f005:**
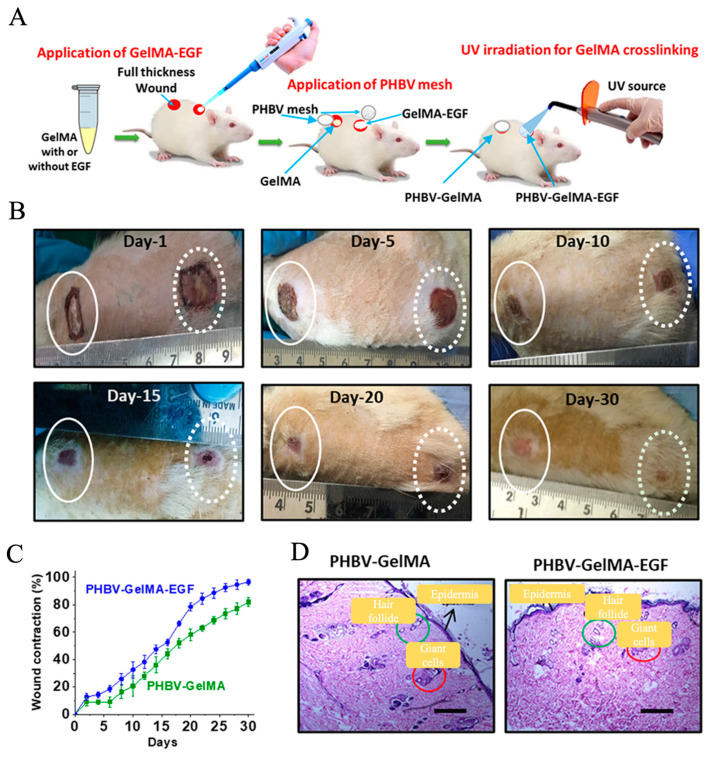
Development and application of electrospun nanofibers/hydrogel composite dressings for the treatment of DUs. (**A**) Schematic illustration of the fabrication and application of composite dressings consisting of electrospun PHBV nanofiber meshes and GelMA hydrogels loading with or without EGF. (**B**) The actual photographs of the diabetic wound areas treated with PHBV-GelMA-EGF composite dressing on days 1, 5, 10, 15, 20, and 30. (**C**) Wound contraction analysis. (**D**) HE stain of wound sites treated with PHBV-GelMA-EGF and PHBV-GelMA composite dressing on day 30. Scale bars = 200 μm. Reprinted with permission from Ref. [167].

**Figure 6 pharmaceutics-15-02285-f006:**
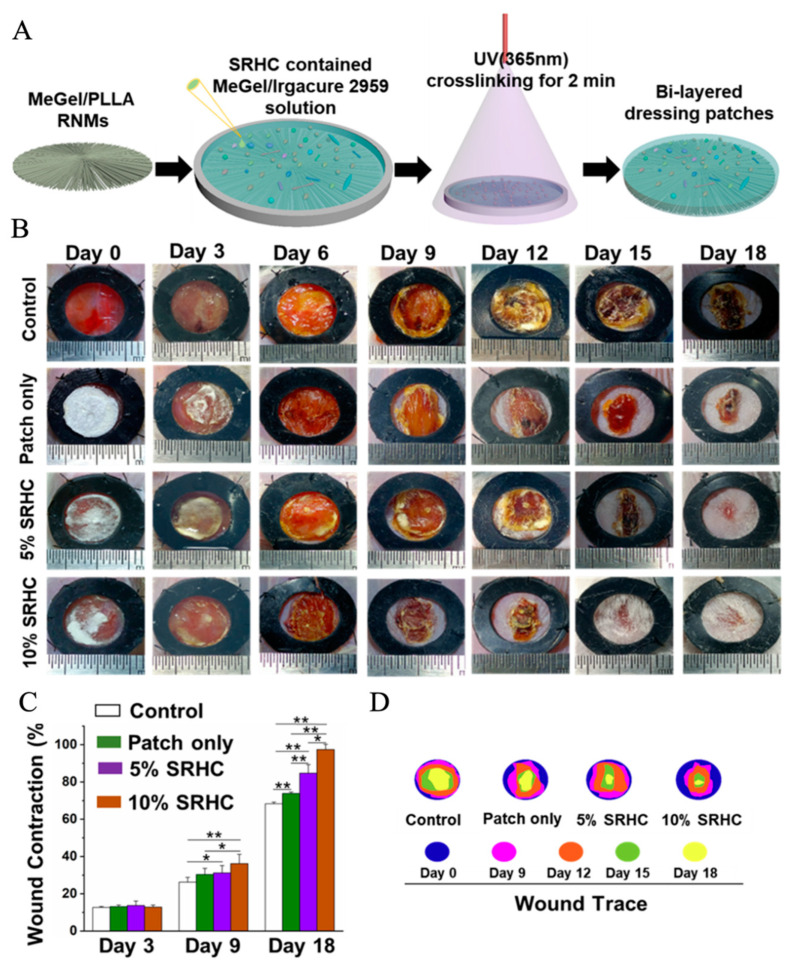
Development and application of bi-layered electrospun nanofibers/hydrogel composite dressing patches for the treatment of DUs. (**A**) Schematic illustration of the fabrication process of the bi-layered composite dressing patches consisting of one layer of MeGel/PLLA RNMs and one layer of SRHC-contained MeGel hydrogels. (**B**) The actual photographs of the diabetic wound areas. (**C**) Wound contraction analysis and (**D**) dynamic display of wound closure treated with composite dressing patches loaded with different concentrations of SRHC on days 0, 3, 6, 9, 12, 15, and 18. * *p* < 0.05, ** *p* < 0.01. Reprinted with permission from Ref. [172].

**Table 1 pharmaceutics-15-02285-t001:** Lists of some representative electrospun multi-polymers for DU application.

Polymers	Solvent	Mean Fiber Diameter (nm)	Tensile Stress (MPa)	Animal Model	Biological Performance	Ref.
PCL/SF	HFIP	453–950	100.5 ± 5.0	6-week-old Kunming mice	Cell proliferation; reduce inflammatory response	[51]
PLGA/Collagen	HFIP	150–650	96 ± 13.0	S.D. rats	Promoting cell proliferation; cell adhesion	[84]
DBC/PLA	TFA	NONE	NONE	Hairless mice	Increasing the skin remodeling for wound healing	[93]
CA/Zein/Sesame	Acetic acid water	150–250	NONE	Normal mice and Diabetic mice	Promoting keratinocyte growth	[102]
Gelatin/BC	Acetic acid/DMF	220–390	NONE	Sprague Dawley rats	Enhanced adhesion; proliferation and neurite extension	[103]
Silk fibroin/Fenugreek	HFIP	309–439	1.90 ± 4.57	Healthy male Wistar rats	Enhancing proliferation; re-epithelialization	[104]
PANI/CS	TFA/DCM	111–160	NONE	Staphylococcus aureus	Anti-inflammatory; antibacterial	[105]
Silk fibroin/Collagen/ PLCL	HFIP	72–162	NONE	NONE	Enhancing adhesion and proliferation; improving hydrophilic	[106]
Chitosan/PEO	90% aqueous acetic acid	250–330	6.4 ± 0.47	NONE	Good potential for wound dressing	[107]
Silk sericin/PVA	Lukewarm water	130–167	NONE	Male mice	Antibacterial and antioxidant potential	[108]
Gelatin/Fibrinogen	HFIP	150–300	0.0125–0.46	Hongkong mouse	Revealing higher cell proliferation	[109]
Zein/Gelatin	HFIP	69–950	NONE	Male mice	Supporting cell adhesion and proliferation	[110]
PLA/Gelatin	HFIP	230–360	NONE	NONE	Supporting the adhesion and proliferation of cells	[111]
Chitosan/PVA	Acetic acid	90–100	NONE	Rabbit Schwann cells	Higher water uptake	[112]
CNTs/PPDO	HFIP	214–216	225 ± 5.1	Rabbits	Promoting cell growth; cell regeneration	[113]

**Table 2 pharmaceutics-15-02285-t002:** Bioactive agents incorporated into nanofibers for improving wound healing.

Polymers	Solvent	Drug	Mean Fiber Diameter (nm)	Animal Model	Biological Performance	Ref.
CA	Acetone and DMF	AgNO_3_	80–130	Swiss albino mice (25–30 g)	Antibacterial properties, antimicrobial activity	[118]
PEO	Mixed ethanol and deionized water	Zein	156–540	Kunming mice	Promoting the wound healing process	[122]
PLATMC	Deionized water	Gel MA/PAA	NONE	STZ-infected diabetic trauma mouse	Antibacterial	[135]
CS/PVA	Deionized water	UA	100–200	Qingdao mice	Anti-inflammation, antioxidation	[136]
PEG-PCL	Methylene chloride	EGF	NONE	Female C57BL/6 mice (14–16 g)	Enhanced keratinocytic expression	[142]
PLCL	Deionized water	Spidroins NTW1-4CT	239–625	Spidroins	Improving the cytocompatibility	[143]
CA/Gelatin	HFIP	ZM	NONE	Male Wistar rats	Biocompatible and antibacterial	[144]
PVA-CMC-PEG	HFIP	NA	NONE	New Zealand rabbits	Biocompatibility	[145]
Silk fibroin/PVA	Water	CP/GSNO	NONE	NONE	Promoting cell proliferation and antibacterial	[146]
PLGA/Collagen	HFIP	Glucophage	203–410	Eighteen Sprague–Dawley rats	Increasing collagen content	[147]
PCL/Collagen	Glacial acetic acid	Melilotus officinalis extract	160–373	Shaved rats	Regenerate tissues of the skin and prevent infections	[148]
PLA/Chitosan	TFA-AA/DMF	Curcumin	NONE	Mice	Anti-inflammatory and antioxidant	[149]
CCS	DI water	Ibuprofen	410–510	NONE	Promote cell migration and proliferation	[150]
PVA	Water	Triterpenes	NONE	NONE	Anti-inflammatory	[151]
CA	DMF/Acetone Water	ZnO/AgNPs	NONE	NONE	Antibacterial	[152]
Chitosan-PVA	Acetic acid	Lignin	350-790	MICE	Biocompatibility and antibacterial	[91]
PLGA	Acetone/Dichloromethane/ DMF	Amoxicillin/ Dopamine	NONE	NONE	Promoting cell proliferation	[153]
SF-PVA	Water	GSNO	80–300	NIH3T3 fibroblast cells	Enhancing blood supply	[154]
PLA/Chitosan	Deionized water	Cod liver oil	50–150	Kerman University of Medical Animal’s farm	Acceleration of the potential mechanisms	[155]
PVA-KGM	Acetic acid	SSD/PG	250–400	Dossy laboratory male mice (C57BL/6J)	Good biocompatibility degradability	[156]

## Data Availability

The data presented in this study are available in this article.

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
