# Peer review of "Recent Advances in Electrospun Nanofiber-Based Strategies for Diabetic Wound Healing Application"

_pharmaceutics, 2023, doi:10.3390/pharmaceutics15092285_

Round 1

Reviewer 1 Report

no comments

Author Response

Please see the attachment-REVIEWER1

Reviewer 2 Report

The Review describes a wide range of materials obtained by electrospinning from synthetic (PCL, PLGA, PVA, PLA and PLCL) and natural polymers (chitosan, collagen, gelatin, zein, elastin, silk and alginate). According to the criterion of fiber size, all the obtained materials demonstrate biomimetic properties, the fibers have a diameter in the nanometer range (from 50 to 1000 nm), which corresponds to the dimension of the components of the extracellular matrix. Despite the fact that these small fibers have a large specific surface area, the delivery and long-term release of drugs with their help can be very effective. For better therapeutic properties, biologically active ingredients, antibiotics, antibacterial particles, stem cells, metal nanoparticles and plant metabolites were introduced into the fibers. Based on the description of the action of the delivered substances, most of them have an antiseptic effect, some also have an anti-inflammatory effect.

The main question that arises regarding the protocol for conducting in vivo experiments to determine the wound-healing effect of materials is which model animals were used. It is advisable to describe this part in more detail in the review.

Usually, rats showing symptoms of type 2 diabetes are used for such studies. These are either rats with spontaneous Torii diabetes (SDT), which is an inbred line of Sprague-Dawley rats, or animals that are modeled with type 2 diabetes by injecting streptozotocin intraperitoneally once at a dose of 65 mg / kg. Then, when applying model wounds to animals, the dynamics of their healing and the nature of the healing processes will be affected by a violation of the activity of cells involved in healing (keratinocytes, fibroblasts, mesenchymal stem cells, vascular cells, immune, etc.). Therefore, for the prevention and treatment of diabetic foot, triggers are needed that regulate the behavior of cells at the level of genetic programs.

The article describes the wound-healing effect of structures based on the fibrous part combined with a hydrogel. They may be able to create an environment in the wound that is necessary for cells to promote cell migration and proliferation, but this will probably not be enough for the treatment of diabetic foot. But as wound coverings that prevent infection of wounds, this is a very interesting group of materials.

Author Response

Please see the attachment-REVIEWER 2

Reviewer 3 Report

This is a review for the Review article discussing the use of Electrospun Nanofibers in treating diabetic wounds.

This was a really nice, comprehensive review. Each section provided large amounts of information, summarizing the research to date. 

The figures and tables added much to the paper too. The schematic illustrations were clear and informative. Where appropriate, tables were provided to summarize the articles published corresponding to the relevant sections.

In table 2., there were a few entries in the Animal Model column that did not specify the animal used. This was also the case with one entry in table one.

Is there any way to make the tables more easy to read? In some areas, the text from one entry is hard to view as it is merging with adjacent entries.

Author Response

Please see the attachment-REVIEWER 3
